# Regulation of H9C2 cell hypertrophy by 14-3-3η via inhibiting glycolysis

Sha Wan[1⊛], Songhao Wang[1⊛], Xianfei Yang[2], Yalan Cui[1,3], Heng Guan[1], Wenping Xiao[1], Fang Liu[1,4]*

1 Department of Anatomy, College of Basic Medicine, Guilin Medical University, Guilin, China, 2 Guangxi Key Laboratory of Brain and Cognitive Neuroscience, Guilin Medical University, Guilin, China, 3 Clinical Pathology Department, The Second People's Hospital of Yichang, Yichang, China, 4 Center of Diabetic Systems Medicine, Guangxi Key Laboratory of Excellence, Guilin Medical University, Guilin, China

⊛ These authors contributed equally to this work.
* fang.liu@msn.com

**Data Availability Statement:** All relevant data are within the manuscript and its Supporting Information files.

## Abstract

It has been reported that Ywhah (14-3-3η) reduces glycolysis. However, it remains unclear about the downstream mechanism by which glycolysis is regulated by 14-3-3η in cardiac hypertrophy. As an important regulator, Yes-associated protein (YAP) interacts with 14-3-3η to participate in the initiation and progression of various diseases in vivo. In this study, the model of H9C2 cardiomyocyte hypertrophy was established by triiodothyronine (T3) or rotenone stimulation to probe into the action mechanism of 14-3-3η. Interestingly, the over-expression of 14-3-3η attenuated T3 or rotenone induced cardiomyocyte hypertrophy and decreased glycolysis in H9C2 cardiomyocytes, whereas the knockdown of 14-3-3η had an opposite effect. Mechanistically, 14-3-3η can reduce the expression level of YAP and bind to it to reduce its nuclear translocation. In addition, changing YAP may affect the expression of lactate dehydrogenase A (LDHA), a glycolysis-related protein. Meanwhile, LDHA is also a possible target for 14-3-3η to mediate glycolysis based on changes in pyruvate, a substrate of LDHA. Collectively, 14-3-3η can suppress cardiomyocyte hypertrophy via decreasing the nucleus translocation of YAP and glycolysis, which indicates that 14-3-3η could be a promising target for inhibiting cardiac hypertrophy.

## Introduction

Cardiac hypertrophy is an adaptive response to hemodynamic stress, which is commonly regarded as a crucial compensatory role to enhance cardiac performance [1, 2]. Various factors such as hypertension or hyperthyroidism can lead to cardiac hypertrophy [3, 4].Persistent cardiac hypertrophy is related to dysfunction and cardiac remodeling, which finally reduces adaptability and increases the risk of heart failure [5, 6].

14-3-3 proteins, gene name is *YWHA*, belong to a highly conserved acidic protein family. It is present in almost all eukaryotic cells, and acts as a "chaperone molecule" or adaptor [7]. The 14-3-3 protein has seven subtypes (α/β, ε, η, γ, σ, θ/τ, and δ/ζ). The 14-3-3η protein, a crucial member of the 14-3-3 protein family, exhibits predominant localization within mitochondria

**Funding:** The funders had no role in study design, data collection and analysis, decision to publish, or preparation of the manuscript.

**Competing interests:** The authors have declared that no competing interests exist.

[8]. The interaction between 14-3-3η and lactate dehydrogenase A (LDHA) inhibits the glycolysis metabolic pathway and promotes mitochondrial biogenesis [9]. Moreno-Vicente's research indicates that the interaction between 14-3-3η and Yes-associated protein (YAP) could regulate the loss of cell YAP activity and the nuclear translocation of YAP [10]. Furthermore, the relationship between YAP and glycolysis is closely linked in hypertrophic cardiomyocytes [11]. These findings suggest that 14-3-3η may exert an important impact on the intracellular glycolysis process via its interaction with LDHA and YAP.

Previous studies suggest that rate of glycolysis was elevated in hypertrophic hearts as compared with control [12–14]. In the absence of oxygen and ischemia, lactate production is also increased in hypertrophic hearts compared with normal ones [15, 16]. In contrast, the activation of glycolysis is strongly associated with cell growth, including cardiac hypertrophy [17, 18]. These data indicate that the glycolytic capacity is closely associated with cardiac hypertrophy.

Our previous study showed that the expression of 14-3-3η was increased in T3-stimulated HL-1 cells and the knockdown of 14-3-3η could aggravate T3-induced cardiac hypertrophy [19]. Here, we found that 14-3-3η could inhibit H9C2 cardiomyocyte hypertrophy by reducing the expression and nuclear translocation of YAP. Moreover, 14-3-3η may mediate H9C2 cardiomyocyte hypertrophy via the inhibition of aerobic glycolysis. This implies that 14-3-3η could be a possible therapeutic target for inhibiting cardiac hypertrophy.

## Materials and methods

### Cell culture

H9C2 cardiomyocytes were provided by Cell Bank of the Chinese Academy of Sciences (Shanghai, China). H9C2 cardiomyocytes were cultured in high glucose Dulbecco's Modified Eagle Medium (C11995500BT, Gibco, NY, USA) supplemented with 10% (v/v) fetal bovine serum (FBS, 10270-106, Gibco, NY, USA), 1% (v/v) penicillin/streptomycin (P1400, Solarbio, Beijing, China). H9C2 cells were placed in an incubator with a constant humidity at 37°C in a 5% carbon dioxide ($CO_2$) atmosphere. For passaging, the cells were rinsed with phosphate-buffered saline (PBS; without calcium and magnesium ions ($Ca^{2+}$ and $Mg^{2+}$, respectively)) and released by pancreatin (T1300, Solarbio, Beijing, China). The replacement of the culture medium was done every other day. The maintenance of the cells followed the protocols of the manufacturer.

### Transfection and treatment

Small interfering ribonucleic acids (siRNAs) were used to knock down 14-3-3η. The sense sequence of siRNA-Ywhah is 5′-GGAGGGUUAUUAGUAGCAUTT-3′, and the antisense sequence is 5′-AUGCUACUAAUAACCCUCCTT-3′. The sense sequence of negative control (siRNA-NC) is 5′-UUCUCCGAACGUGUCACGUTT-3′, and the antisense sequence is 5′-ACGUGACACGUUCGGAGAATT-3′. H9C2 cells were transfected with siRNA-Ywhah or siR-NA-NC (GenePharma, Suzhou, China) by Lipofectamine 3000 reagent (L3000-015, Invitrogen, CA, USA) for 48 hours. The Ywhah overexpression plasmid (CMV promoter, Genechem, Shanghai, China) or its empty control vector were transfected into H9C2 cardiomyocytes by Lipofectamine 3000 reagent for 48 hours. According to previous reports [20–22], 1000 nM of triiodothyronine (T3) (T2877, Sigma-Aldrich, St. Louis, MO, USA) or 100 nM of rotenone (R8875, Sigma-Aldrich, St. Louis, MO, USA) were utilized for the 48 hours stimulation of H9C2 cells. The H9C2 cells were treated with T3 and 10 μM Verteporfin [23] (SML0534, Sigma-Aldrich, St. Louis, MO, USA) for 48 hours to establish a YAP-inhibited model of hypertrophy.

## Immunofluorescence staining

After being cultured and plated in 6-well plates, H9C2 cells were rinsed with PBS, fixed in 4% polyformaldehyde and permeabilized with 0.3% TritonX-100 for 20 minutes. The samples were blocked with 5% BSA (#A8020, Solarbio, Beijing, China) in PBST at room temperature for 1 hour, and then incubated with primary antibodies against YAP (66900-1-Ig, Proteintech, Wuhan, China, 1:100 dilution) overnight at 4°C. Fluorescein isothiocyanate (FITC)-labeled secondary antibodies against mouse (ZF-0513, ZSGB-BIO, Beijing, China) were used to visualize the signals. Then, 4', 6-diamidino-2-phenylindole (DAPI, Solarbio, Beijing, China) was used for counterstaining the nucleus and visualized under a fluorescence microscope (Olympus, Japan). All images were further processed by the ImageJ software.

## Western blot analysis

A Radio Immunoprecipitation Assay (RIPA) buffer containing protease inhibitors was used for the 10-minute lysis of H9C2 cells on ice. A bicinchoninic acid assay (BCA) analysis kit (P0010, Beyotime, Shanghai, China) was utilized for the detection of protein concentrations. An electrophoresis apparatus of 10% sodium dodecyl sulphate-polyacrylamide gels (SDS-PAGE) was loaded with an equal amount of protein extraction (30 μg). After electrophoresis was completed, the transfer of the blot onto a polyvinylidene fluoride (PVDF) membrane (Millipore) was followed by timed incubation with the chosen mouse against YAP (66900-1-Ig, Proteintech, Wuhan, China), Histone H3 (68345-1-Ig, Proteintech, Wuhan, China), and rabbit against LDHA (3558S, Cell Signaling Technology, Boston, USA), 14-3-3η (ab206292, Abcam, Cambridge, UK), actin (#4970, Cell Signaling Technology, Boston, USA) primary antibodies at 4 °C overnight. Subsequently, it underwent 1-hour incubation with secondary antibodies goat anti-rabbit or anti-mouse IgG-horseradish peroxidase (HRP) (SA00001-2, Proteintech, Wuhan, China, or ZB-2305, ZSGB-BIO, Beijing, China) at room temperature. At last, an enhanced chemiluminescence (ECL) (P10100, NCM Biotech, Suzhou, China) detection system was adopted to detect the bands.

## Co-immunoprecipitation (co-IP)

H9C2 cells were collected and washed with PBS. After 15 min of lysis, the supernatant was incubated with 5ug Ywhah antibodies at 4°C with rotation overnight. 50 μL Protein A/G PLUS-Agarose beads (sc-2003, Santa Cruz Biotechnology, CA, USA) were added, incubated with rotation for 4 hours at 4°C, and then centrifuged at 12,000×g for 1 min. Next, washed the beads three times with 1 mL of lysis buffer and then boiled in loading buffer. The samples were subjected to Western blot analysis.

## Wheat germ agglutinin staining

Cells were stained with wheat germ agglutinin (25530, WGA, AAT Bioquest, CA, USA) for the measurement of cell surface area. In the late phase of the treatment, 4% paraformaldehyde was used to fix cells for 25 minutes, then stained with WGA for 30 minutes. After that, DAPI was used for the 10-minute staining of the nucleus. A laser confocal microscope (Nikon, Japan) was employed to observe cells. Eventually, cell images from no less than five randomly selected fields were analyzed to measure cell surface area by use of Image J software.

## Lactate level measurement

A lactate assay kit (A019-2-1, Nanjing Jiancheng Bioengineering Institute, Nanjing, China) was utilized for detecting lactate content in the cell culture medium as per the instructions of the manufacturer.

## Intracellular adenosine triphosphate and pyruvate content in H9C2 cells

Adenosine triphosphate (ATP) (S0026, Beyotime, Shanghai, China) and pyruvate assay kits (BC2205, Solarbio, Beijing, China) were used for determining ATP and pyruvate in cell lysates following the instructions of the manufacturer. Before testing, the cells were evenly distributed on a 24-well plate or 60 mm Petri dish according to the number of cells required by the kit. Cells were sown at 30,000 cells/well in a 24-effewell plate and 100,000 cells/well in a 60 mm plate for 24 hours.

## Cytoplasmic and nuclear components separation

Cytoplasmic and nuclear extracts were prepared according to the instructions of the nuclear and cytoplasmic extraction kit (P0027, Beyotime, Shanghai, China).

## Detection of protein/DNA ratio

Protein concentration in H9C2 cell lysates were quantified by Protein concentration assay kit (P0010S, Beyotime, Shanghai, China). And the concentration of DNA in cell lysates was quantified by Hoechst33342 staining solution (C0031, Solarbio, Beijing, China) under fluorescence intensity of multiscan spectrum at an excitation wavelength of 350 nm and an emission wavelength of 461 nm.

## Detection of cell viability

Cell counting kit-8 (CCK-8) assay (#HY-K0301, MedChemExpress, Shanghai, China) was utilized for test H9C2 cell viability. Briefly, cells were cultivated into 96-well plate (5000 cells/well) for 12 hours at 37°C and the cells were treated as above mentioned method. Then, 10 μL kit solution was mixed into the culture medium for 90 minutes. A Microplate Sepctrophotometer was used to determine the absorbance at 450 nm.

## Quantitative RT-PCR (qRT-PCR)

TRIzol reagent (15596026, Thermo Fisher, St. Louis, MO, USA) was used to extract total RNA from cultured H9C2 cells. The cDNA was reverse transcribed as the protocol by using Rever-Tra Ace® qPCR RT Master Mix with gDNA Remover (FSQ-301, TOYOBO, Osaka, Japan). Power Up SYBR Green PCR Master Mix (A25742, Thermo Fisher, St. Louis, MO, USA) was used to perform quantitative RT-PCR and used GAPDH as an internal control. The relative expression levels of the target genes of interest were calculated using the $2^{-\Delta\Delta Ct}$ method. The following primer sequences were used: ANP forward primer is 5′-GGAAGTCAACCCGTCTC AGA-3′, and reverse primer is 5′-TGGGCTCCAATCCTGTCAAT-3′; β-MHC forward primer is 5′-CCAGTCCCGAGGTGTACTTT-3′, and reverse primer is 5′-TCCTCCTTCATGTTGGCC AT-3′; GAPDH forward primer is 5′-GACATGCCGCCTGGAGAAAC-3′, and reverse primer is 5′-AGCCCAGGATGCCCTTTAGT-3′.

## Statistical analysis

All data were indicated by mean ± standard deviation (SD). Statistical analysis was performed using GraphPad Prism 8.0.2 (GraphPad Software, Inc., San Diego, CA, USA). The analysis of statistical differences was conducted by a one-way analysis of variance (ANOVA) with a least significant difference (LSD) multiple comparisons test. It was considered that $P<0.05$ showed statistical significance.

## Results

### Inhibition of triiodothyronine-induced cardiomyocyte hypertrophy by 14-3-3η via inhibiting glycolysis

To understand the role of 14-3-3η protein in glycolytic metabolism and T3-induced cardiomyocyte hypertrophy, the related marker of glycolysis and hypertrophy were detected under overexpression of 14-3-3η in H9C2 cells. After the overexpression of 14-3-3η, T3 was used for the 48 hours treatment of H9C2 cells (**Fig 1A and 1B**). The result showed that the level of lactate exhibited a significant increase in T3-stimulated H9C2 cells compared with the control, this effect was counteracted after the overexpression of 14-3-3η (**Fig 1D**). This suggests that 14-3-3η may inhibit the aerobic glycolysis of hypertrophic cardiomyocytes. In the meantime, previous research mentioned that the effects of 14-3-3η on glycolysis may be caused by the inhibition of LDHA [9]. After T3 stimulation, LDHA expression showed a remarkable upregulation. however, this augmentation was diminished in 14-3-3η overexpressed cells (**Fig 1A and 1C**). To further verify if 14-3-3η can affect glycolysis by inhibiting LDHA, the intracellular content of pyruvate, a substrate of LDHA, was detected. Consistently, the intracellular content of pyruvate further increased in comparison with the empty control after the overexpression of 14-3-3η in T3 treatment cells (**Fig 1E**). The above results suggested that 14-3-3η inhibits the glycolysis in T3-stimulated hypertrophic cardiomyocytes by inhibiting LDHA.

The mRNA expression levels of ANP and β-MHC, biomarkers of cardiac hypertrophy, were upregulated in T3-induced H9C2 cells compared with the control, and this trend was reverted by the overexpression of 14-3-3η (**Fig 1F**). The cell surface area was significantly increased in T3-stimulated H9C2 cells as detected by WGA staining, but this effect was counteracted by overexpression of 14-3-3η (**Fig 1G and 1H**). In addition, the ratio of protein/DNA, a marker of cellular hypertrophy, was remarkably increased upon T3 stimulation, and this augmentation could be diminished by overexpression of 14-3-3η (**Fig 1I**). These results indicated that 14-3-3η could attenuated T3-induced cardiomyocyte hypertrophy by inhibiting glycolysis.

### Inhibition of rotenone-induced cardiomyocyte hypertrophy by 14-3-3η via inhibiting glycolysis

Previous study show that cardiomyocytes are converted from oxidative phosphorylation to glycolysis with thyroid hormone treatment [24]. To further explore the role of glycolysis on cardiomyocyte hypertrophy, we treated H9C2 cells with rotenone, which can block the oxidative respiratory chain and rapidly increases the glycolysis activity [25]. After the overexpression of 14-3-3η, rotenone was used for the 48 hours treatment of H9C2 cells (**S1A Fig**). Rotenone treatment markedly increased lactate content in the culture medium, and this trend was weakened by the elevation of 14-3-3η (**S1B Fig**). The production of ATP was remarkably reduced in rotenone-treated cells in comparison with the control, but it showed no significant difference after the overexpression of 14-3-3η (**S1C Fig**). Meanwhile, the expression of LDHA was significantly enhanced with rotenone treatment, and this increasing effect was attenuated by the overexpression of 14-3-3η (**S1D Fig**). In addition, this phenomenon persisted when T3 and rotenone were used together (**S1F Fig**). Moreover, intracellular pyruvate showed a significant increase in rotenone stimulation compared with the control, and pyruvate was further accumulated after the overexpression of 14-3-3η compared with the empty control (**S1E Fig**). Taken together, these data indicate that 14-3-3η inhibits the glycolysis of rotenone-induced H9C2 cardiomyocyte hypertrophy, and this effect is possibly attributable to LDHA, a key protein during glycolysis.

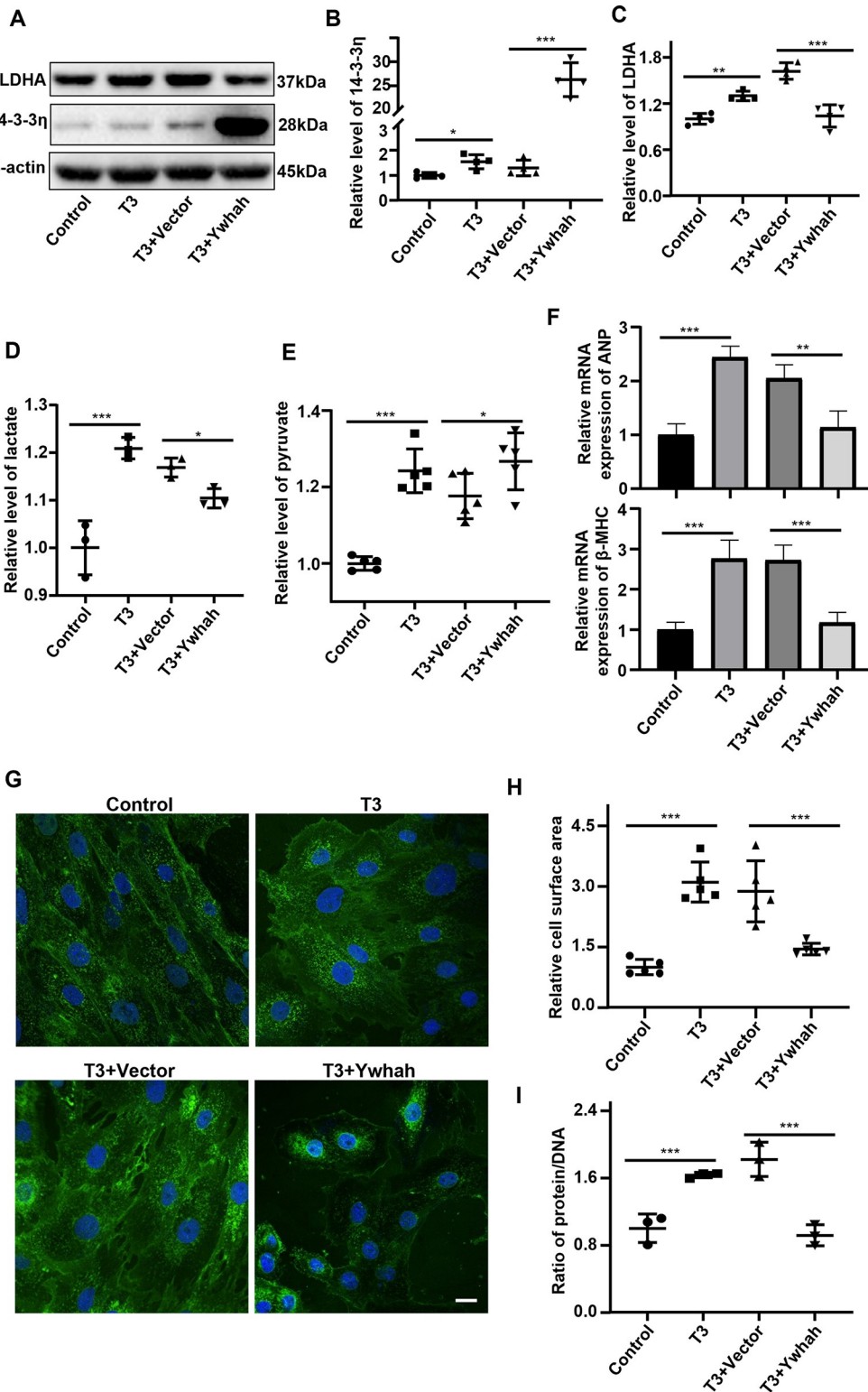

**Fig 1. Inhibition of triiodothyronine-induced cardiomyocyte hypertrophy by 14-3-3η via inhibiting glycolysis.**
H9C2 cells were transfected with Ywhah or empty Vector plasmid for 8 hours and then stimulated with 1000 nM T3
for 48 hours. **A.** Representative Western blot images show the level of Ywhah and LDHA protein. **B.** Semi-quantitative
analysis of Ywhah protein detected by Western Blot. **C.** Semi-quantitative analysis of LDHA protein detected by
Western Blot. **D.** Lactate, the product of glycolysis, was detected in the cell culture supernatant. **E.** Pyruvate, the

product of glycolysis, was detected in cell lysate. **F.** The mRNA level of ANP and β-MHC detected by RT-qPCR. **G.** and **H.** Representative WGA staining images (**G**) show the cell surface area and their statistical analysis (**H**). Bar = 10 μm. **I.** Protein/DNA ratio show hypertrophy. Data were analyzed by one-way analysis of variance [ANOVA] with LSD posttest (* $P<0.05$, ** $P<0.01$, *** $P<0.001$), each symbol in graphs B-E and I representing an independent experiment, each symbol in graph G representing a random microscopic field.

WGA staining showed that cardiomyocyte surface area was remarkably increased with rotenone treatment, and this increasing effect was strongly diminished by the overexpression of 14-3-3η (**S2A and S2B Fig**). On balance, 14-3-3η inhibits rotenone-induced H9C2 cardiomyocyte hypertrophy by inhibiting glycolysis. To detect whether 14-3-3η influence the viability of H9C2 cells, we performed CCK-8 assay. T3 stimulation shows no obvious impact on the cell viability (**S3A Fig**). Due to rotenone can block the respiratory chain complex 1 in mitochondrion, the cell viability was decreased with rotenone stimulation (**S3B Fig**). Overexpression of 14-3-3η has no effect on the viability of H9C2 cells compared with the vector control (**S3A and S3B Fig**). These results indicate that 14-3-3η do not influence the cell viability.

## Promotion of glycolysis by the knockdown of 14-3-3η to aggravate triiodothyronine-induced cardiomyocyte hypertrophy

Above results indicate that 14-3-3η can inhibit glycolysis to affect cardiomyocyte hypertrophy. Then, is the knockdown of 14-3-3η is sufficient to drive cardiomyocyte growth? Further research was carried out by the knockdown of 14-3-3η with siRNAs in H9C2 cells. The siRNA-Ywhah could obviously knock down 14-3-3η which detected by Western blot (**Fig 2A**). In subsequent experiments, 14-3-3η expression was silenced by siRNA transfection, followed by T3 treatment (**Fig 2B**). The lactate content in the culture medium demonstrated a significant increase after T3 stimulation, and it could be further increased by the knockdown of 14-3-3η (**Fig 2C**). Importantly, LDHA was significantly elevated after T3 stimulation. Then, knockdown of 14-3-3η alone was sufficient to increase LDHA expression in T3-induced H9C2 cells (**Fig 2D**). Moreover, intracellular pyruvate was markedly up-regulated after T3 stimulation, but the accumulation of pyruvate was reduced after the knockdown of 14-3-3η (**Fig 2E**). This result suggests that the knockdown of 14-3-3η increases LDHA, which leads to the decrease of pyruvate accumulation. In conclusion, the level of glycolysis in T3-induced hypertrophic cardiomyocyte can be exacerbated after knockdown of 14-3-3η.

To explore the effect of 14-3-3η on cardiomyocyte hypertrophy, WGA staining was performed to verify the change of surface area in cardiomyocytes after the knockdown of 14-3-3η. The surface area of cardiomyocytes exhibited a remarkable increase compared with the control by T3 treatment, which was further aggravated after the knockdown of 14-3-3η (**Fig 2F and 2G**). Interestingly, the ratio of protein/DNA was significantly elevated in T3-induced H9C2 cells, and knockdown of 14-3-3η can reinforce this increased effect (**Fig 2H**). The data proved that knocking down 14-3-3η exacerbated T3-induced H9C2 cardiomyocyte hypertrophy by inhibiting glycolysis.

## Promotion of glycolysis by the knockdown of 14-3-3η to aggravate rotenone-induced cardiomyocyte hypertrophy

14-3-3η was silenced by siRNAs in H9C2 cells and then stimulated with rotenone for 48 hours (**S4A Fig**). Lactate was detected. The lactate content in the culture medium was increased by rotenone treatment and further increased by the knockdown of 14-3-3η (**S4B Fig**). Moreover, the ATP detection results verified that intracellular ATP content was drastically reduced under rotenone stimulation, but it was significantly increased by the knockdown of 14-3-3η compared with the NC control (**S4C Fig**). In addition, LDHA expression showed a remarkable

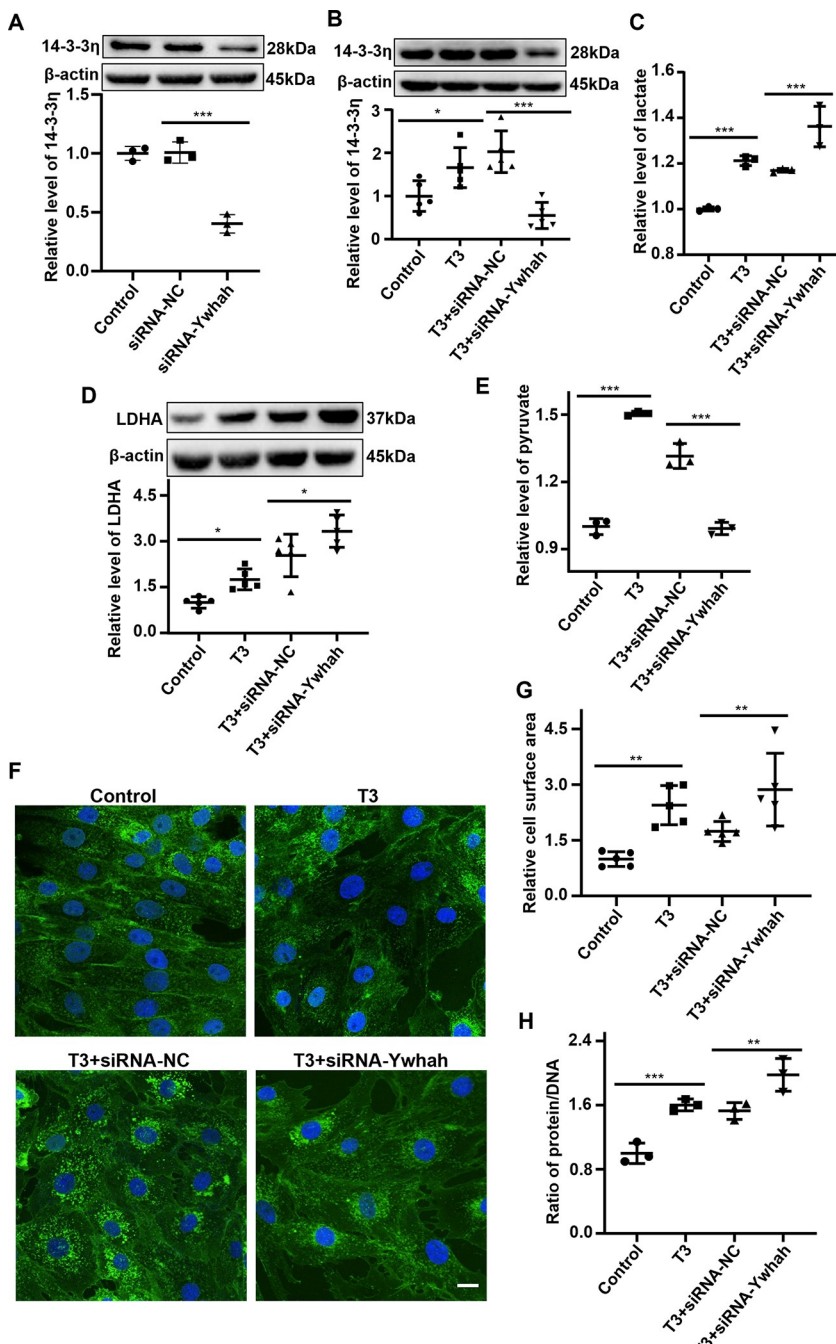

**Fig 2. Promotion of glycolysis by the knockdown of 14-3-3η to aggravate triiodothyronine-induced cardiomyocyte hypertrophy. A.** H9C2 cells were transfected with siRNA-Ywhah or NC for 48 hours, the knockdown efficiency of 14-3-3η was detected by Western blot. **(B-G)** H9C2 cells were transfected with siRNA-Ywhah or NC for 8 hours and then stimulated with 1000 nM T3 for 48 hours. **B.** Representative Western blot and semi-quantification statistical data showing the expression of 14-3-3η protein. **C.** Lactate, the product of glycolysis, was detected in the cell culture supernatant. **D.** Representative Western blot and semi-quantification statistical data showing the expression of LDHA, a glycolysis-related protein. **E.** Pyruvate, the product of glycolysis, was detected in cell lysate. **F.** and **G.** Representative WGA staining images (**F**) show the cell surface area and their statistical analysis (**G**). Bar = 10 μm. **H.** Protein/DNA ratio show hypertrophy. Data were analyzed by one-way analysis of variance [ANOVA] with LSD posttest (* $P<0.05$, ** $P<0.01$, *** $P<0.001$), each symbol in graphs A-E and H representing an independent experiment, each symbol in graph G representing a random microscopic field.

upregulation after rotenone stimulation and then a further increase after the knockdown of 14-3-3η (**S4D Fig**). The same phenomenon occurred when H9C2 cells were stimulated by T3 and rotenone simultaneously (**S4F Fig**). After rotenone stimulation, pyruvate showed a significant upregulation. This augmentation was, however, diminished in 14-3-3η knockdown cells (**S4E Fig**). To sum up, these findings suggest that the knockdown of 14-3-3η can exacerbate the glycolysis of rotenone-stimulated H9C2 cardiomyocyte hypertrophy by acting on LDHA.

WGA staining showed that the surface area of cardiomyocytes exhibited a significant increase compared with the control under rotenone treatment, and a further increase when 14-3-3η was inhibited (**S5A and S5B Fig**). The above experimental results show that the inhibition of 14-3-3η promotes glycolysis, which leads to the exacerbation of cardiomyocyte hypertrophy caused by rotenone stimulation.

## Inhibition of glycolysis and H9C2 cardiomyocyte hypertrophy by 14-3-3η via regulating YAP

YAP is a transcription factor witch can initiate a variety of physiological processes. Its localization in cells is also relevant to how it functions [26]. Previous study shows that 14-3-3η and YAP can interact within cells, which can change the activity and localization of YAP and ultimately affect the function of YAP [10]. To address this issue, the interaction of YAP and 14-3-3η was examined in H9C2 cardiomyocytes by co-immunoprecipitation (**Fig 3A**). The results showed that YAP can directly binds to 14-3-3η in H9C2 cardiomyocytes. Subsequently, H9C2 cardiomyocytes were over-expressed by 14-3-3η and then stimulated with T3, rotenone or both for 48 hours. The Western blot results revealed that the expression of YAP showed a marked increase under the influence of T3 or rotenone, while the increase in YAP was significantly attenuated after the overexpression of 14-3-3η (**Fig 3B–3D**). To understand the intracellular localization of YAP, immunofluorescence staining experiments and a fluorescence brightness analysis of nuclear-to-cytoplasmic ratios were performed. It was found that a large amount of YAP was retained in the cytoplasm after the overexpression of 14-3-3η under T3 (**Fig 4A**) or rotenone (**S6A and S6B Fig**) stimulation. Meanwhile, the level of YAP was increased in the cytoplasm and was decreased in the nucleus after overexpression of Ywhah under T3 stimulation (**Fig 4B**). The results of this study indicate the interaction of 14-3-3η with YAP, in which 14-3-3η inhibits the expression of YAP and reduces the nuclear translocation of YAP in H9C2 cells.

## Promotion of glycolysis and H9C2 cardiomyocyte hypertrophy by the knockdown of 14-3-3η via regulating YAP

The expression of YAP was significantly increased in H9C2 cardiomyocytes after the stimulation of T3 or rotenone. However, it was further aggravated after the knockdown of 14-3-3η (**Fig 3E–3G**). Immunofluorescence staining revealed that knocking down 14-3-3η aggravated the expression of YAP entering the nucleus with T3 (**Fig 5A**) or rotenone (**S7A and S7B Fig**) stimulation. In addition, the level of YAP was decreased in the cytoplasm and was increased in the nucleus after silencing Ywhah in T3 treated H9C2 cells (**Fig 5B**). Collectively, these findings suggest that the forced reduction of 14-3-3η exacerbates the expression and nuclear translocation of YAP in H9C2 cardiomyocytes by T3 or rotenone treatment.

## Expression of the glycolysis-associated protein LDHA suppressed by the inhibition of YAP

YAP participates in the glycolysis of tumor cells and plays a vital role in tumor growth and migration [27]. However, it is still unclear how the alteration of YAP plays a role in

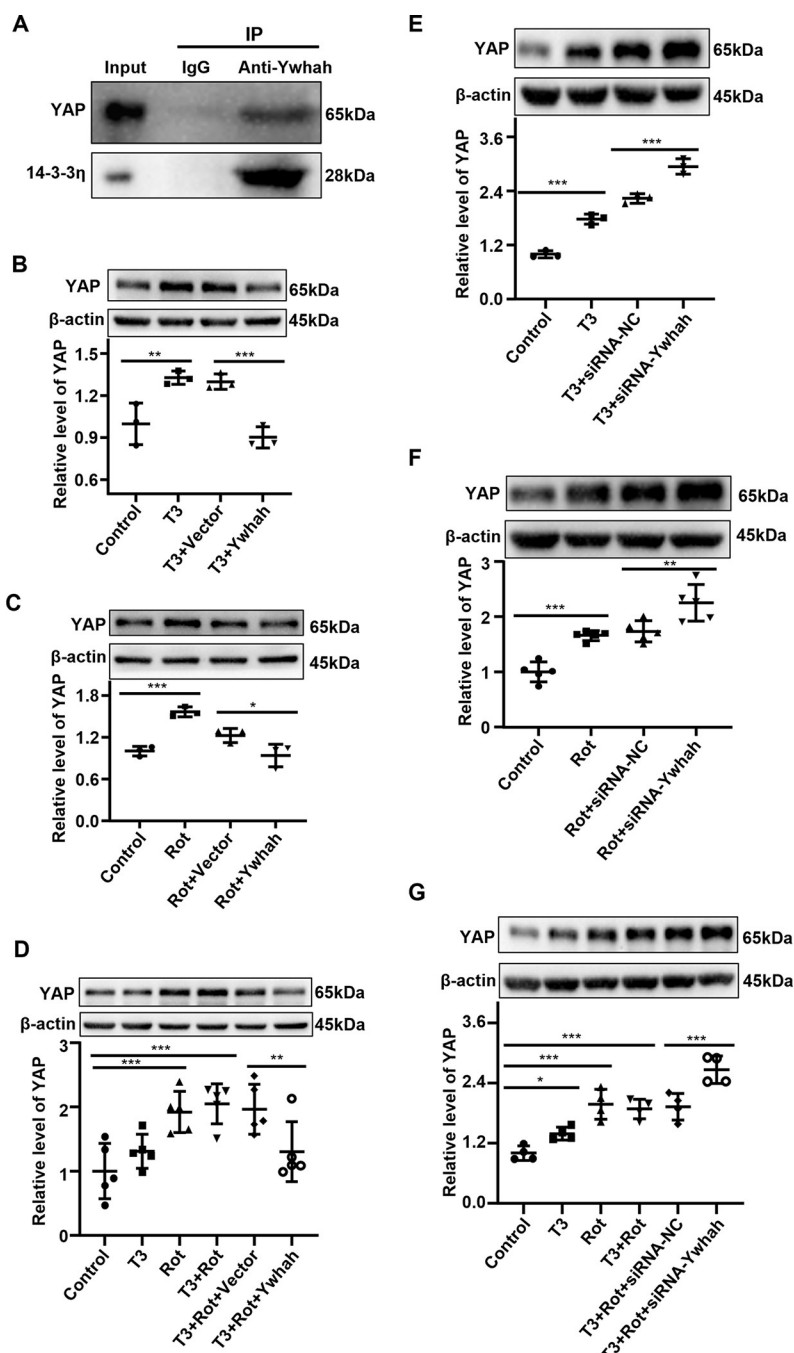

**Fig 3. The expression of YAP was regulated by 14-3-3η. A.** Co-immunoprecipation shows that YAP interacts with 14-3-3η in H9C2 cells. **(B-D)** H9C2 cells were transfected with Ywhah or empty Vector plasmid for 8 hours, the expression of YAP was detected by Western blot after further 48 hours stimulation with T3 (**B**), rotenone (Rot) (**C**) or co-stimulation (**D**). **(E-G)** H9C2 cells were transfected with siRNA-Ywhah or NC for 8 hours, the expression of YAP was detected by Western blot after further 48 hours stimulation with T3 (**E**), rotenone (**F**) or co-stimulation (**G**). Data were analyzed by one-way analysis of variance [ANOVA] with LSD posttest (* $P<0.05$, ** $P<0.01$, *** $P<0.001$), each symbol in graphs B-G representing an independent experiment.

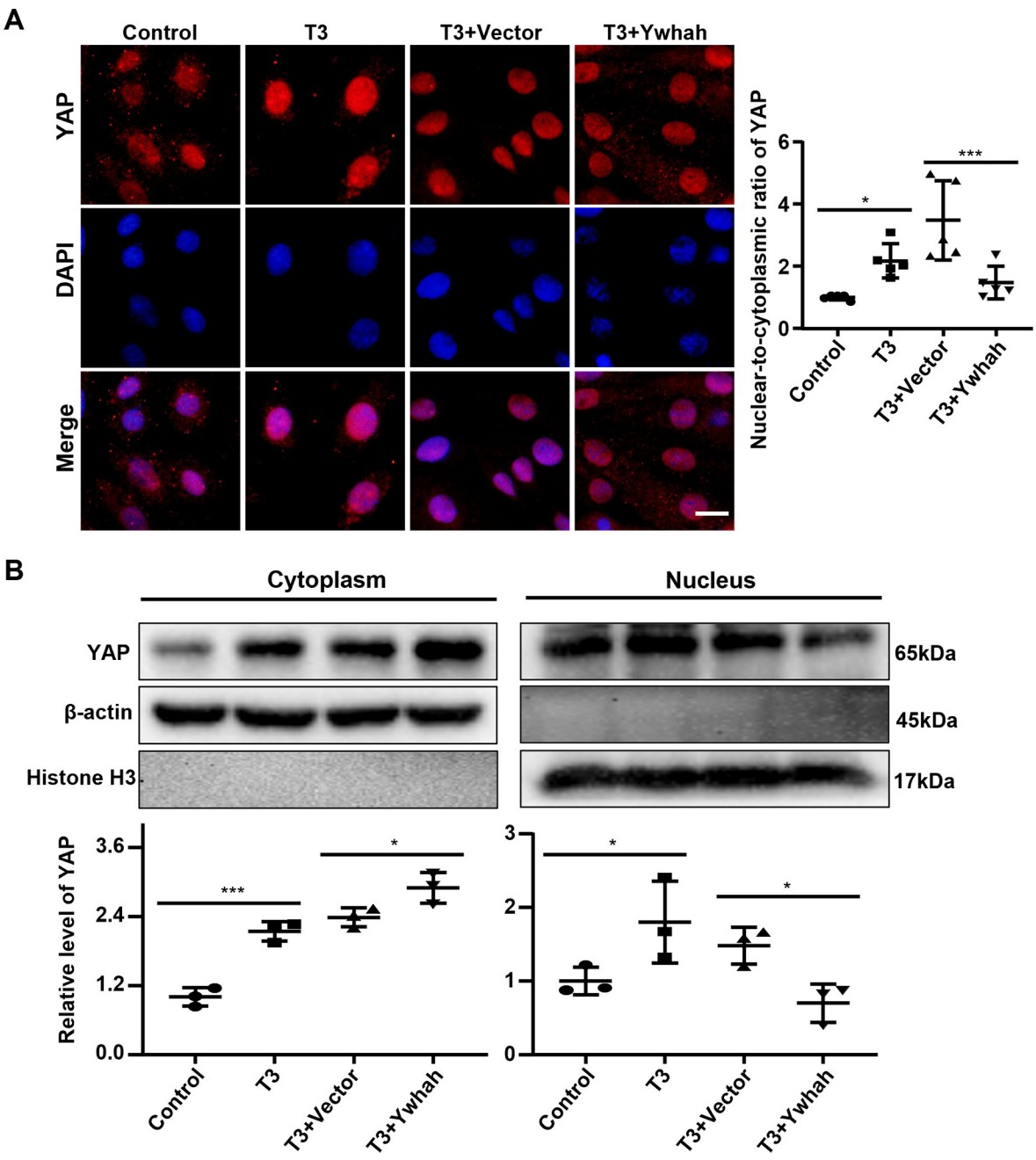

**Fig 4. Overexpression of 14-3-3η could suppress triiodothyronine-induced YAP nuclear translocation.** H9C2 cells were transfected with Ywhah or empty Vector plasmid for 8 hours, the nuclear translocation of YAP was detected by immunofluorescence staining (**A**) and the expression level was detected by Western blot (**B**) after further 48 hours stimulation with T3. Bar = 20 μm. Data were analyzed by one-way analysis of variance [ANOVA] with LSD posttest (* $P<0.05$, *** $P<0.001$), each symbol in graph A representing a random microscopic field, each symbol in graphs B representing an independent experiment.

hyperthyroid cardiac hypertrophy and whether it influences the glycolysis of H9C2 cardiomyocyte hypertrophy. To explain these doubts, it is necessary to clarify the downstream molecules of YAP action. Research reported that verteporfin suppresses cell survival by disrupting the YAP-transcriptional enhancer factor domain (TEAD) complex [28]. In the present study, H9C2 cardiomyocytes were stimulated for 48 hours with verteporfin to inhibit the effect of

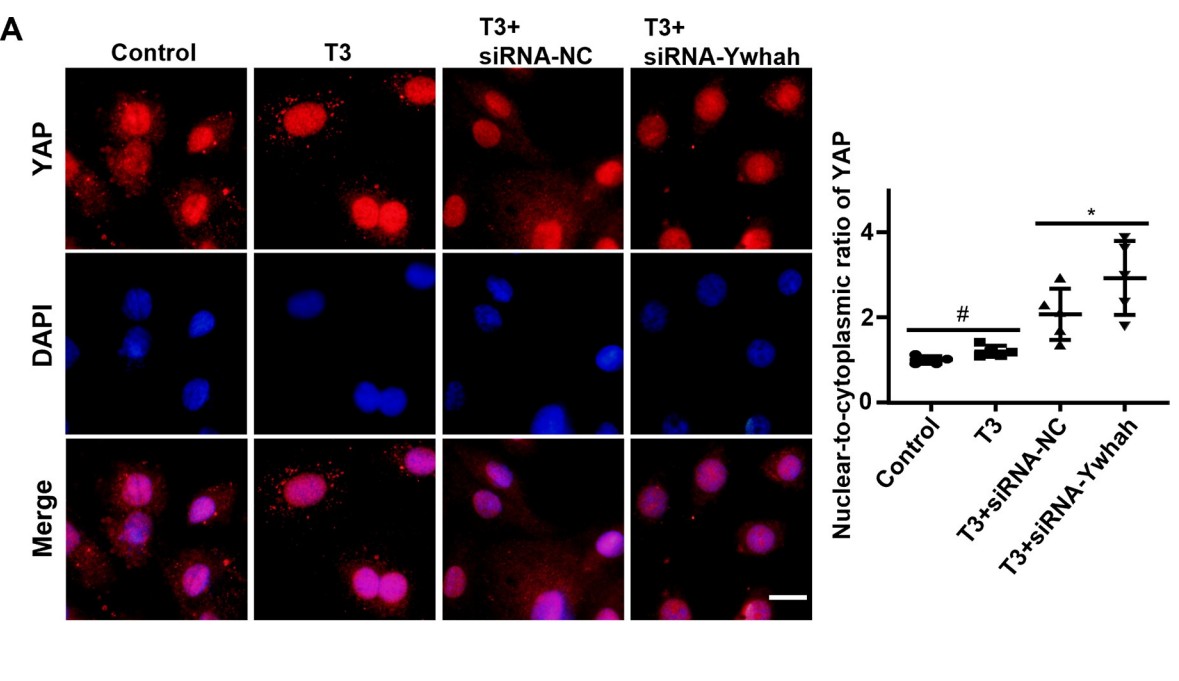

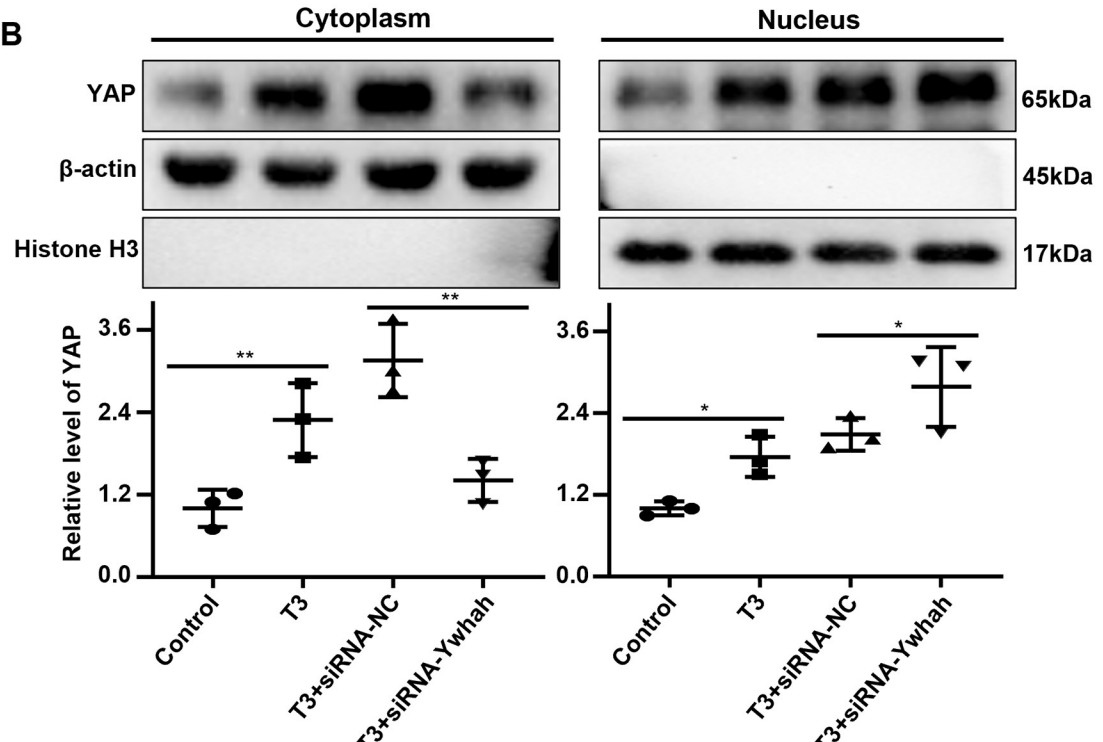

**Fig 5. Knockdown of 14-3-3η could increase triiodothyronine-induced YAP nuclear translocation.** H9C2 cells were transfected with siRNA-Ywhah or NC for 8 hours, the location of YAP was detected by immunofluorescence staining (**A**) and the expression level was detected by Western blot (**B**) after further 48 hours stimulation with T3. Bar = 20 μm. Data were analyzed by one-way analysis of variance [ANOVA] with LSD posttest (* $P<0.05$, ** $P<0.01$), each symbol in graph A representing a random microscopic field, each symbol in graphs B representing an independent experiment. # $P<0.05$ by student $t$-test.

YAP. After YAP was inhibited by verteporfin in T3-stimulated H9C2 cells, the protein level of LDHA was reduced (**Fig 6A and 6B**), and this reduction also occurred in rotenone-stimulated H9C2 cells (**Fig 6C–6E**). Overall, these results show that inhibiting YAP can reduce the expression of the glycolysis-related protein LDHA with the stimulation of T3 or rotenone.

## Discussion

Hyperthyroidism accelerates basal metabolic rates and oxidative metabolism by inducing specific mitochondrial enzymes. This increases the production of reactive oxygen species (ROS) and decreases anti-oxidant capacity, which leads to cardiac hypertrophy [29]. Continued hypertrophic growth of the heart may ultimately result in heart failure and sudden death [30, 31]. Understanding the molecular mechanism of pathological cardiac hypertrophy will be beneficial for the treatment of heart failure. Up to now, studies have discovered various proteins that regulate cardiac hypertrophy, like YAP and LDHA [11, 32]. Controlling these proteins can alleviate the pathological process of cardiac hypertrophy to some degree and lower the probability of adverse cardiac events.

As a member of the YWHA protein family, 14-3-3η has been studied for its extensive involvement in inflammation, apoptosis, and intracellular energy metabolism [33–35]. In recent years, studies have found that the 14-3-3η protein can inhibit glycolysis through the interactions between LDHA and YAP [9, 10]. In present study, glycolysis was increased in T3-stimulated H9C2 cardiomyocytes, which is consistent with the previous study showing that thyroid hormone can increase glycolysis in C2C12 cells [24]. However, this effect was reduced after the overexpression of 14-3-3η. Additionally, the change in ATP exhibited no statistical significance in the same situation. The level of oxidative phosphorylation in hyperthyroid H9C2 cells is unknown, and oxidative phosphorylation is an important way to generate ATP. Hence, ATP cannot be used as a criterion to evaluate the glycolysis level in the hyperthyroid cell model.

Mitochondria generate ~95% of the ATP in normal heart, and they are regarded as the central organelles which are responsible for the coordination of energy [36]. A recent study show that compensated cardiac hypertrophy is characterized by a parallel change of cardiac mass and mitochondrial content and function [37]. Mitochondria play a key role in cell metabolism, and they have a significant cross-talk with the other component in the cytoplasm. NADH, a cytosolic reducing equivalents which formed during glycolysis, are oxidized by mitochondria [38]. Our previous study indicated that 14-3-3η protein could regulate thyroxine-induced mitochondrial damage and mitophagy in cardiomyocytes [19]. Therefore, we speculate that 14-3-3η protein may regulate mitochondrial function to affect the process of glycolysis.

To further verify the association between 14-3-3η and glycolysis, rotenone was selected for experiments. Respiratory chain complex 1 was blocked in H9C2 cardiomyocytes by rotenone, which led to a sharp increase in the rate of glycolysis [25]. After the overexpression of 14-3-3η, the content of intracellular ATP showed no difference in rotenone-stimulated cells. It was suspected that rotenone blocked the oxidative respiratory chain and glycolysis was used to produce energy for survival. Therefore, the inhibitory effect of 14-3-3η is not remarkable. In conclusion, the detection of glycolysis-related products and protein expression suggests that the overexpression of 14-3-3η inhibits the glycolysis process. In this study, the role of 14-3-3η in the glycolysis of cardiomyocytes was revealed.

As an adaptive compensatory mechanism, cardiac hypertrophy maintains cardiac output during the process of harmful stimuli. However, long-term irritation can trigger chronic hypertrophy and may lead to heart failure [39]. Our previous study found that overexpression of the 14-3-3η ameliorated T3-induced HL-1 cardiomyocyte hypertrophy, whereas

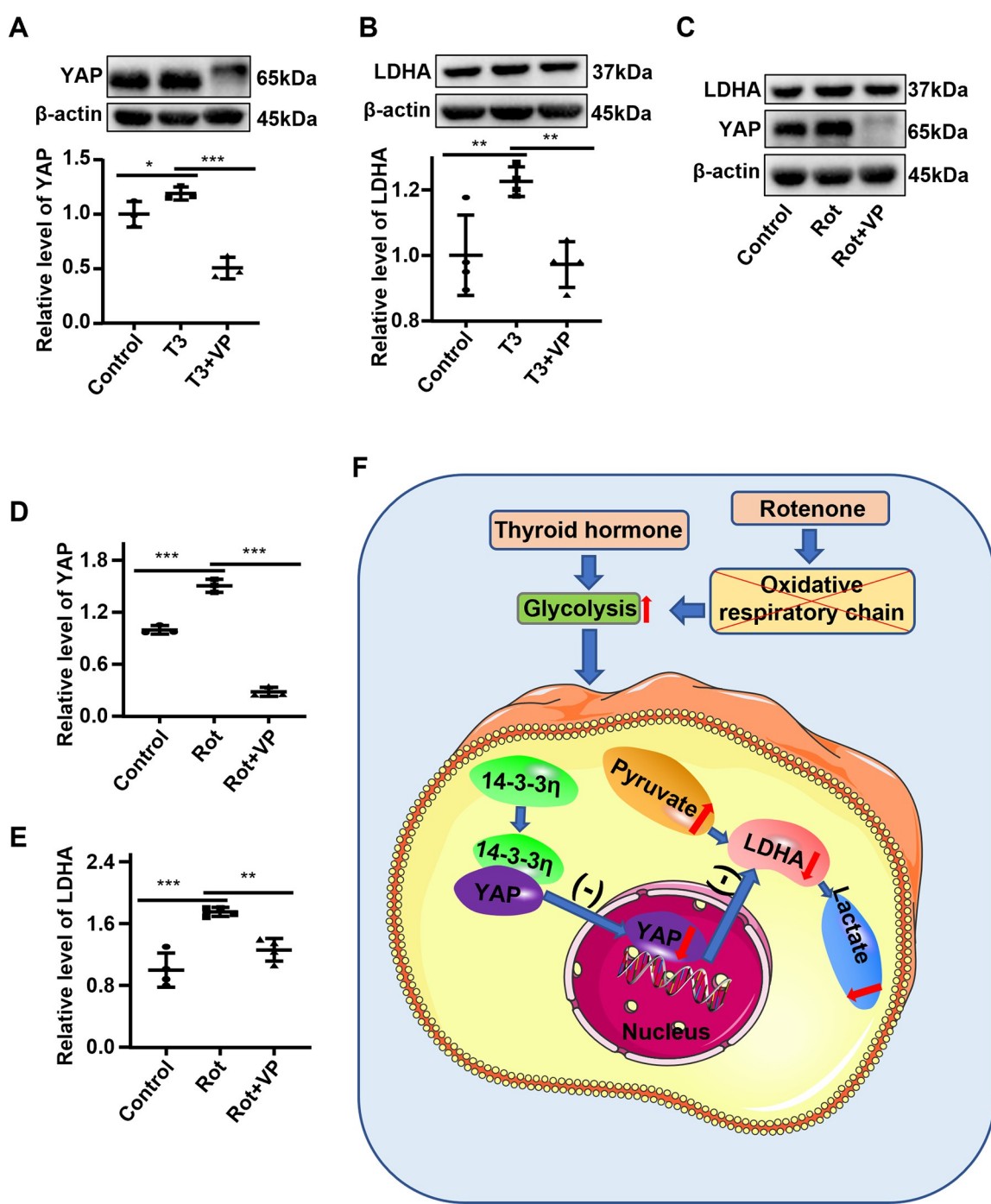

**Fig 6. The expression of LDHA, a glycolysis-associated protein, was suppressed by the inhibition of YAP. (A-B)** H9C2 cells were treated with T3 and verteporfin (VP) for 48 hours. Representative Western blot and semi-quantification statistical data showing the expression of YAP (**A**) and LDHA (**B**) protein. **(C-E)** H9C2 cells were treated with rotenone (Rot) and verteporfin (VP) for 48 hours. Representative Western blot **(C)** and semi-quantification statistical data showing the expression of YAP **(D)** and LDHA **(E)** protein. **F.** The graphical summary of the function and mechanism of 14-3-3η in H9C2 cells. Data were analyzed by one-way analysis of variance [ANOVA] with LSD posttest (* $P<0.05$, ** $P<0.01$, *** $P<0.001$), each symbol in graphs A, B, D, and E representing an independent experiment.

knockdown of the 14-3-3η protein aggravated T3-induced HL-1 cardiomyocyte hypertrophy [19]. Therefore, T3 or rotenone was used to stimulate H9C2 cells in the current experiment. Then, WGA staining found that the overexpression of 14-3-3η can reduce H9C2 cardiomyocyte hypertrophy caused by thyroid hormone or rotenone stimulation.

The Hippo pathway and its downstream effectors, namely the transcriptional coactivator YAP and that with PDZ-binding motifs (TAZ), regulate the growth of organs and the plasticity of cells during the development and regeneration of animals [40]. Research shows that YAP plays a role in entering the nucleus when the myocardium is exposed to pressure overload, which causes compensatory cardiac hypertrophy and avoids the occurrence of heart failure [11]. In addition, 14-3-3η interacts with YAP to mediate the cytoplasmic retention of YAP, which inhibits the occurrence of cardiac hypertrophy [10]. Subsequently, the interaction between 14-3-3η and YAP was verified in H9C2 cells in this study. Specifically, YAP can inhibit cardiac hypertrophy by inhibiting glycolysis. Moreover, the nuclear translocation of YAP is closely bound up with cardiac hypertrophy [11]. Following this line of thought, this study investigated the role played by 14-3-3η in mediating glycolysis and cardiac hypertrophy through YAP. The possible mechanism of the role of 14-3-3η in compensatory cardiac hypertrophy was elucidated.

This study has some limitations. Although 14-3-3η was found to inhibit cardiomyocyte hypertrophy through glycolysis, whether it is also involved in other mechanisms to influence cardiac hypertrophy is unclear. Moreover, 14-3-3η interacts with YAP to influence glycolysis. Whether 14-3-3η affects glycolysis through other factors is unknown. In this research, it was shown that 14-3-3η can inhibit hyperthyroid cardiomyocyte hypertrophy through glycolysis with the accumulation of intermediates and metabolites of the glycolytic pathway, which may be attributed to its interaction with YAP. Overall, 14-3-3η and glycolysis may be promising targets to inhibit cardiac hypertrophy to some extent while avoiding the further deterioration of cardiac function and adverse cardiac events.

## Supporting information

**S1 Fig. Overexpression of 14-3-3η could suppress rotenone-induced glycolysis.** H9C2 cells were transfected with Ywhah or empty Vector plasmid for 8 hours and then stimulated with 100 nM Rotenone (Rot) for 48 hours. **A.** Representative western blot and semi-quantification statistical data showing the expression of 14-3-3η protein. ## $P<0.01$ by student $t$-test. **B.** Lactate, the product of glycolysis, was detected in the cell culture supernatant. **C.** ATP level was detected in cell lysate. **D.** Representative western blot and semi-quantification statistical data showing the expression of LDHA, a glycolysis-related protein. **E.** Pyruvate, the product of glycolysis, was detected in cell lysate. **F.** Representative western blot and semi-quantification statistical data showing the level of LDHA in Ywhah or empty Vector plasmid transfected H9C2 cells undergo co-stimulation with T3 and Rotenone (Rot). Data were analyzed by one-way analysis of variance [ANOVA] with LSD posttest (\* $P<0.05$, \*\* $P<0.01$, \*\*\* $P<0.001$, NS, no significant), each symbol in graphs A-F representing an independent experiment.
(TIF)

**S2 Fig. Overexpression of 14-3-3η can attenuate rotenone-induced cardiomyocyte hypertrophy.** H9C2 cells were transfected with Ywhah or empty Vector plasmid for 8 hours and then stimulated with 100 nM Rotenone (Rot) for 48 hours. **A.** and **B.** Representative WGA staining images (**A**) show the cell surface area and their statistical analysis (**B**). Bar = 10 μm. Data were analyzed by one-way analysis of variance [ANOVA] with LSD posttest (\*\* $P<0.01$, \*\*\* $P<0.001$), each symbol representing a random microscopic field.
(TIF)

**S3 Fig. Overexpression of 14-3-3η has no effect on the viability of H9C2 cells.** (**A-B**) H9C2 cells were transfected with Ywhah or empty Vector plasmid for 8 hours, the cell viability was detected by CCK-8 assay after further 48 hours stimulation with T3 (**A**) or rotenone (Rot) (**B**). Data were analyzed by one-way analysis of variance [ANOVA] with LSD posttest (*** $P<0.001$, NS, no significant).
(TIF)

**S4 Fig. Knockdown of 14-3-3η could aggravate rotenone-induced glycolysis.** H9C2 cells were transfected with siRNA-Ywhah or NC for 8 hours and then stimulated with 100 nM Rotenone (Rot) for 48 hours. **A.** Representative western blot and semi-quantification statistical data showing the expression of 14-3-3η protein. **B.** Lactate, the product of glycolysis, was detected in the cell culture supernatant. **C.** ATP level was detected in cell lysate. **D.** Representative western blot and semi-quantification statistical data showing the expression of LDHA, a glycolysis-related protein. **E.** Pyruvate, the product of glycolysis, was detected in cell lysate. **F.** Representative western blot and semi-quantification statistical data showing the level of LDHA in siRNA-Ywhah or NC transfected H9C2 cells undergo co-stimulation with T3 and Rotenone (Rot). Data were analyzed by one-way analysis of variance [ANOVA] with LSD posttest (* $P<0.05$, ** $P<0.01$, *** $P<0.001$), each symbol in graphs A-F representing an independent experiment.
(TIF)

**S5 Fig. Knockdown of 14-3-3η can enhance rotenone-induced cardiomyocyte hypertrophy.** H9C2 cells were transfected with siRNA-Ywhah or NC for 8 hours and then stimulated with 100 nM Rotenone (Rot) for 48 hours. **A.** and **B.** Representative WGA staining images (**A**) show the cell surface area and their statistical analysis (**B**). Bar = 10 μm. Data were analyzed by one-way analysis of variance [ANOVA] with LSD posttest (* $P<0.05$, *** $P<0.001$), each symbol representing a random microscopic field.
(TIF)

**S6 Fig. Overexpression of 14-3-3η can attenuate rotenone-induced YAP translocation from cytoplasm to nucleus.** H9C2 cells were transfected with Ywhah or empty Vector plasmid for 8 hours and then stimulated with 100 nM Rotenone (Rot) for 48 hours. **A.** and **B.** Detection the nuclear translocation of YAP by immunofluorescence staining (**A**) and their statistical analysis (**B**). Bar = 20 μm. Data were analyzed by one-way analysis of variance [ANOVA] with LSD posttest (* $P<0.05$, *** $P<0.001$), each symbol representing a random microscopic field.
(TIF)

**S7 Fig. Knockdown of 14-3-3η can enhance rotenone-induced YAP translocation from cytoplasm to nucleus.** H9C2 cells were transfected with siRNA-Ywhah or NC for 8 hours and then stimulated with 100 nM Rotenone (Rot) for 48 hours. **A.** and **B.** Detection the nuclear translocation of YAP by immunofluorescence staining (**A**) and their statistical analysis (**B**). Bar = 20 μm. Data were analyzed by one-way analysis of variance [ANOVA] with LSD posttest (*** $P<0.001$), each symbol representing a random microscopic field. ### $P<0.001$ by student *t*-test.
(TIF)

**S1 Raw image. Raw Western blot images.**
(PDF)

## Author Contributions

**Conceptualization:** Sha Wan, Fang Liu.

**Data curation:** Songhao Wang, Fang Liu.

**Formal analysis:** Sha Wan, Fang Liu.

**Funding acquisition:** Fang Liu.

**Investigation:** Sha Wan, Songhao Wang, Fang Liu.

**Methodology:** Sha Wan, Songhao Wang, Xianfei Yang, Yalan Cui, Heng Guan, Fang Liu.

**Project administration:** Fang Liu.

**Resources:** Sha Wan.

**Software:** Xianfei Yang.

**Supervision:** Fang Liu.

**Writing – original draft:** Sha Wan, Xianfei Yang, Fang Liu.

**Writing – review & editing:** Sha Wan, Songhao Wang, Xianfei Yang, Yalan Cui, Wenping Xiao, Fang Liu.

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
