## [Decision Letter · Decision Letter 0]

26 Apr 2024

PONE-D-24-06553Regulation of H9C2 cell hypertrophy by 14-3-3η via inhibiting glycolysisPLOS ONE

Dear Dr. Liu,

Thank you for submitting your manuscript to PLOS ONE. After careful consideration, we feel that it has merit but does not fully meet PLOS ONE’s publication criteria as it currently stands. Therefore, we invite you to submit a revised version of the manuscript that addresses the points raised during the review process.

We look forward to receiving your revised manuscript.

Kind regards,

Vincenzo Lionetti, M.D., PhD

Academic Editor

PLOS ONE

Journal Requirements:

"National Natural Science Foundation of China (82160063)"

**Additional Editor Comments:**

**ACADEMIC EDITOR: **All issues addressed by Reviewers are required.However, the authors should pay particular attention to mitochondrial function. Indeed, interference of Warburg effect improves mitochondrial function and cardiac function in the process of cardiac hypertrophy and heart failure. 3,5,3'-Levo-triiodothyronine (T3) increases the expression of factors involved in mitochondrial DNA transcription and biogenesis, such as hypoxic inducible factor-1α, mitochondrial transcription factor A and peroxisome proliferator activated receptor γ coactivator-1α much more in the presence of oxidative microenvironment through mitoKATP dependent pathway (J Cell Mol Med. 2011 Mar;15(3):514-24). The authors should investigate the role of these mitochondrial factors in mediating 14-3-3η  effects.

Reviewers' comments:

Reviewer's Responses to Questions

**Comments to the Author**

1. Is the manuscript technically sound, and do the data support the conclusions?

Reviewer #1: Partly

Reviewer #2: Yes

2. Has the statistical analysis been performed appropriately and rigorously? 

Reviewer #1: Yes

Reviewer #2: Yes

3. Have the authors made all data underlying the findings in their manuscript fully available?

Reviewer #1: No

Reviewer #2: Yes

4. Is the manuscript presented in an intelligible fashion and written in standard English?

Reviewer #1: Yes

Reviewer #2: No

5. Review Comments to the Author

Reviewer #1: Comments to the Authors

The article “Regulation of H9C2 cell hypertrophy by 14-3-3η via inhibiting glycolysis”, from Wan and colleagues, investigates the role of Ywhah (14-3-3η) in the regulation of cardiomyocytes hypertrophy through inhibition of glycolysis. Through modulation of 14-3-3η expression, the authors observe how the protein regulates cardiomyocyte cell size and glycolysis, the latter by determining a reduced LDHA expression, which they demonstrate could happen via downregulation of YAP expression and nuclear translocation. The study is generally well designed and the rationale behind the analyses is clear. Methods and results are appropriately described. The paper readability is satisfactory and the figures are acceptable, however the manuscript could benefit from a revision by a native English speaker, as some errors were spotted throughout the text.

I have only a few issues that I believe would be useful for paper publication, followed by some minor comments to improve paper clarity.

Main comments

1. How were the dosage of T3 and Rotenone chosen? From previous studies? In that case, I would add a reference.

2. In order to talk about hypertrophy, It would be important to combine cell size measurement with an evaluation of cardiomyocytes hypertrophy markers such as ANP, BNP and β-myosin heavy chain/ α-myosin heavy chain.

3. Please add details on the Ywhah plasmid.

4. Please add Verteporfin treatment details in the material and methods section.

Minor comments

1. Line 46: are you sure it is only a systolic dysfunction?

2. Line 62: I am not sure about the article [10] cited, as it is from 1965 and I could not fin a proper mention of 14-3-3.

3. Line 69: I am not sure about ref [14] here, I think ref [13] is more appropriate for this sentence.

4. Line 73: Again this is not the best reference as it does not really mention an effect of 14-3-3η on HIF-1α. Moreover, from literature I could see that actually stabilizes HIF-1α. [Qiu Y, et al., 2019 doi: 10.1038/s41420-019-0200-8]

5. Line 90-92: The sentence need a reference (work on HL1 cells).

6. Line 118: Employed to transfect.

7. Line 119: Was conducted.

8. Line 131: It is better to write secondary antibodies against mouse.

9. Line 164: I think there is an error in this dilution: are you sure it is 2 mg in 500 uL?

10. Line 204: I am not sure about “upstream product”, Isn’t it “substrate” better?

11. Line 208: Are you sure about confirm? Is it better suggest?

12. Line 268: “enforce” is not the best choice of word here, did you mean “reinforce”?

13. Line 355-357: I am not sure ref [14]is adequate for both sentences. Maybe [15] in the first and [13] in the second?

14. Line 377-380: This sentence needs a reference.

15. Do you have better pictures of WGA staining? Maybe a single z-level acquired at a confocal microscope would help having a more defined cell border instead of a diffuse colour. However, the staining is enough to calculate cell area, hence I have no problem with the data obtained in terms of cell size.

16. In figure legends, please use capital letter for Western blot.

Reviewer #2: In this article, the Authors provide a corpus of evidence that Yhwah, a protein of the 14-3-3 family, counteracts the effects of T3 and rotenone on cardiac cell hypertrophy reducing glycolysis by interacting with Yes-associated protein (YAP). To provide such evidence, they showed that:

(i) overexpression of Yhwah reduced the expression of lactate dehydrogenase A (LDHA) and increased the intracellular content of pyruvate.

(ii) overexpression of Yhwah leads to a reduction of cardiomyocyte surface area

(iii) The knockdown of Yhwah gene exacerbates T3-induced myocardial hypertrophy.

(iv) YAP is significantly attenuated after the overexpression of Yhwah

(v) Inhibition of YAP reduces the level of LDHA

The results are sound and the methods appear, in general well suited.

I have some major concerns:

A. The use of rotenone as a model for cardiac hypertrophy. The authors found a significant increase in surface area in cells treated with rotenone, compared to the control group. However, as expected, rotenone reduced ATP content in treated cells (line 277). Can Authors explain how a treatment that leads to a reduction in ATP can determine cell hypertrophy?

B. The study lacks results regarding the viability of cells after various treatments. In other words: in order to conclude that the observed values reflect true differences in the content of single cells, Authors should provide evidence that cells in different conditions have comparable viability.

C. Although I am not a native English speaker, the language needs to be edited throughout the whole manuscript as some sentences (some of them are highlighted in the "minor concerns" part) are difficult to read, or have a wrong syntax.

Furthermore, I have these minor concerns:

1. Introduction should be re-organized:

a1. What does it mean that "cardiac hypertrophy ... helps maintain heart function in its ORIGINAL STAGE" (lines 42-44)? a2. The subsequent parts (i.e. lines 44-46, lines 54-58) are vague and do not add much context.

a3. "Hence, it is a feasible approach ..." (line 58), may be Authors intended "It is worthwhile"?

a4. The corpus of text from line 61 to line 79 consists of single sentences interleaved with full stops, which makes the manuscript hard to read. Please, rephrase in a more organic way.

a5. I would not cite the Warburg effect since it is a bit off-topic for this article (or at least the sentence should be rephrased)

a6. "All in all, glycolysis..." (line 88): please rephrase.

The main point here is to re-organize, and in my opinion, shorten, the introduction regarding cardiac hypertrophy (since there are many articles that describe this topic in detail)

2. Materials and methods

2a. "1% streptomycin/penicillin from solarbio... and 1% antibiotics (Solarbio,..." Please re-check the sentences. Do you put other antibiotics in the medium? Please consider to write the manufacturer's name in the same way throughout the manuscript.

2b. "Mouse against YAP..."(line 129): please rephrase; please cite the antibody that you have used.

3. Results

3a. Line 191-195: please rephrase, as the sentences are difficult to read.

3b. In lines 118-120 Authors said that they conducted three independent bio-repeats, each of which contained three technical replicates: I would expect three different points in the graphs in Figure A. However, the graphs in Figure A have different points in the different conditions (3-5 points per condition). Could the Authors explain in detail how the data that are shown was obtained?

3c. Lines 300-301: "The results confirmed the combination of YAP and 14-3-3n in H9C2 cardiomyocytes.". What does it mean? Please, rephrase.

3d. Fig 1, 3: Please clarify "Vector" in the caption. Do you mean negative control (transfection with empty plasmid?)

6. PLOS authors have the option to publish the peer review history of their article (what does this mean?). If published, this will include your full peer review and any attached files.

Reviewer #1: **Yes: **Giulia Furini

Reviewer #2: **Yes: **Lorenzo Fontanelli

---

## [Author Response · Author response to Decision Letter 0]

9 Jun 2024

Response to Editor’s and Reviewer’s Comments

We thank the editor and reviewers for their insightful reviews and excellent suggestions. We have made changes to our manuscript according to the reviewers’ comments. The following is a point by point response to the comments.

Comments from the Editors:

Comment: However, the authors should pay particular attention to mitochondrial function. Indeed, interference of Warburg effect improves mitochondrial function and cardiac function in the process of cardiac hypertrophy and heart failure. 3,5,3'-Levo-triiodothyronine (T3) increases the expression of factors involved in mitochondrial DNA transcription and biogenesis, such as hypoxic inducible factor-1α, mitochondrial transcription factor A and peroxisome proliferator activated receptor γ coactivator-1α much more in the presence of oxidative microenvironment through mitoKATP dependent pathway (J Cell Mol Med. 2011 Mar;15(3):514-24). The authors should investigate the role of these mitochondrial factors in mediating 14-3-3η effects.

Response: We thank you for the valuable suggestion to improve our manuscript. As per reviewer’s suggestion, we have discussed the relationship between mitochondria and 14-3-3η in the section of discussion in the revised manuscript. Please see line 378-387. 

Reviewer #1: 

The article “Regulation of H9C2 cell hypertrophy by 14-3-3η via inhibiting glycolysis”, from Wan and colleagues, investigates the role of Ywhah (14-3-3η) in the regulation of cardiomyocytes hypertrophy through inhibition of glycolysis. Through modulation of 14-3-3η expression, the authors observe how the protein regulates cardiomyocyte cell size and glycolysis, the latter by determining a reduced LDHA expression, which they demonstrate could happen via downregulation of YAP expression and nuclear translocation. The study is generally well designed and the rationale behind the analyses is clear. Methods and results are appropriately described. The paper readability is satisfactory and the figures are acceptable, however the manuscript could benefit from a revision by a native English speaker, as some errors were spotted throughout the text. I have only a few issues that I believe would be useful for paper publication, followed by some minor comments to improve paper clarity.

Response: We thank reviewer for positive comment. As per reviewer’s suggestion, the revised manuscript has been corrected by a native English speaker.

Major Comments

Comment 1: How were the dosage of T3 and Rotenone chosen? From previous studies? In that case, I would add a reference.

Response: The dosage of T3 and rotenone were from previous studies. We have cited these studies in the revised manuscript. Please see line 98.

Comment 2: In order to talk about hypertrophy, It would be important to combine cell size measurement with an evaluation of cardiomyocytes hypertrophy markers such as ANP, BNP and β-myosin heavy chain/ α-myosin heavy chain.

Response: We have detected the mRNA expression level of ANP and β-myosin heavy chain (β-MHC) by RT-qPCR. Please see Figure 1F and describe in line 216-218.

Comment 3: Please add details on the Ywhah plasmid.

Response: We have added detailed information of the Ywhah plasmid in the revised manuscript. Please see line 96-98.

Comment 4: Please add Verteporfin treatment details in the material and methods section.

Response: We have described Verteporfin treatment in detail in the section of material and methods in the revised manuscript. Please see line 101-103.

Minor Comments

Comment 1: Line 46: are you sure it is only a systolic dysfunction?

Response: We are sorry for the inaccurate description. We have removed this sentence in the revised manuscript.

Comment 2: Line 62: I am not sure about the article [10] cited, as it is from 1965 and I could not fin a proper mention of 14-3-3.

Response: We have removed the ref [10] in the revised manuscript. Please see line 49-50.

Comment 3: Line 69: I am not sure about ref [14] here, I think ref [13] is more appropriate for this sentence.

Response: Yes, ref [13] is more appropriate for this sentence. We have re-cited reference in the revised manuscript. Please see line 53-55.

Comment 4: Line 73: Again this is not the best reference as it does not really mention an effect of 14-3-3η on HIF-1α. Moreover, from literature I could see that actually stabilizes HIF-1α. [Qiu Y, et al., 2019 doi: 10.1038/s41420-019-0200-8]

Response: We have removed this sentence in the revised manuscript. 

Comment 5:Line 90-92: The sentence need a reference (work on HL1 cells).

Response: We have cited a study in the revised manuscript. Please see line 67-69.

Comment 6:Line 118: Employed to transfect.

Response: We have rephrased the sentence in the revised manuscript. Please see line 89-103.

Comment 7: Line 119: Was conducted.

Response: We have rephrased these sentences in the revised manuscript. Please see line 89-103.

Comment 8: Line 131: It is better to write secondary antibodies against mouse.

Response: We have replaced “mouse secondary antibodies” with “secondary antibodies against mouse” in the revised manuscript. Please see line 110.

Comment 9: Line 164: I think there is an error in this dilution: are you sure it is 2 mg in 500 uL?

Response: We are sorry for this mistake, We have removed the sentence in the revised manuscript. 

Comment 10: Line 204: I am not sure about “upstream product”, Isn’t it “substrate” better?

Response: We have replaced “Upstream product” with “substrate” in the revised manuscript. Please see line 35 and 211.

Comment 11: Line 208: Are you sure about confirm? Is it better suggest?

Response: We have replaced “Confirm” with “suggest” in the revised manuscript. Please see line 214.

Comment 12: Line 268: “enforce” is not the best choice of word here, did you mean “reinforce”?

Response: Thanks for the suggestion. We have replaced “Enforce” with “reinforce” in the revised manuscript. Please see line 280.

Comment 13: Line 355-357: I am not sure ref [14]is adequate for both sentences. Maybe [15] in the first and [13] in the second?

Response: We have re-cited these studies in the revised manuscript . Please see line 366-369.

Comment 14: Line 377-380: This sentence needs a reference.

Response: We have cited a study in the revised manuscript. Please see line 400-403.

Comment 15: Do you have better pictures of WGA staining? Maybe a single z-level acquired at a confocal microscope would help having a more defined cell border instead of a diffuse colour. However, the staining is enough to calculate cell area, hence I have no problem with the data obtained in terms of cell size.

Response: We thank you for the valuable suggestion to improve our data. But we regret that we didn’t consider the Z-level acquired at a confocal microscope in that time.

Comment 16: In figure legends, please use capital letter for Western blot.

Response: We have corrected these typing error. Please see Figure legends.

Reviewer #2: 

Comment: In this article, the Authors provide a corpus of evidence that Yhwah, a protein of the 14-3-3 family, counteracts the effects of T3 and rotenone on cardiac cell hypertrophy reducing glycolysis by interacting with Yes-associated protein (YAP). To provide such evidence, they showed that:

(i) overexpression of Yhwah reduced the expression of lactate dehydrogenase A (LDHA) and increased the intracellular content of pyruvate.

(ii) overexpression of Yhwah leads to a reduction of cardiomyocyte surface area

(iii) The knockdown of Yhwah gene exacerbates T3-induced myocardial hypertrophy.

(iv) YAP is significantly attenuated after the overexpression of Yhwah

(v) Inhibition of YAP reduces the level of LDHA

The results are sound and the methods appear, in general well suited.

Response: We thank reviewer for the positive comment. 

Major Comments

Comment 1: The use of rotenone as a model for cardiac hypertrophy. The authors found a significant increase in surface area in cells treated with rotenone, compared to the control group. However, as expected, rotenone reduced ATP content in treated cells (line 277). Can Authors explain how a treatment that leads to a reduction in ATP can determine cell hypertrophy?

Response: The heart relies mainly on mitochondrial metabolism to provide the energy needed for pumping blood. Rotenone, an inhibitor of mitochondrial complex I, reduced the ratio of active mitochondria to total mitochondria, increased ROS production, and decreased ATP production. Studies have revealed alterations in mitochondrial bioenergetic parameters, reporting decreases in oxygen consumption and increases in glycolysis after rotenone exposure (Dranka et al. 2012, Journal of Neurochemistry 122, 941-951. [PubMed: 22708893]; Giordano et al. 2012, PLOS ONE 7, e44610 [PubMed: 22970265]; Karlsson et al. 2016, Mitochondrion 31, 56-62. [PubMed: 27769952]). The increase in glycolysis is, however, accompanied by reduced or normal glucose oxidation, which may lead to an uncoupling between glucose uptake and oxidation. This imbalance has been implicated in pathological hypertrophic remodeling in the heart (Leong et al. 2003, Comp Biochem Physiol A Mol Integr Physiol. 2003;135:499-513). In present study, rotenone induced myocardial hypertrophy maybe cause by elevation of glycolysis.

Comment 2: The study lacks results regarding the viability of cells after various treatments. In other words: in order to conclude that the observed values reflect true differences in the content of single cells, Authors should provide evidence that cells in different conditions have comparable viability.

Response: We have detected the cell viability by CCK-8. The result indicates that 14-3-3η do not influence the cell viability. Please see Supplemental Figure 3 and describe in line 249-255. 

Comment 3: Although I am not a native English speaker, the language needs to be edited throughout the whole manuscript as some sentences (some of them are highlighted in the "minor concerns" part) are difficult to read, or have a wrong syntax.

Response: We have edited throughout the whole manuscript by native English speaker in the revised manuscript.

Minor Comments

Comment 1: Introduction should be re-organized:

a1. What does it mean that "cardiac hypertrophy ... helps maintain heart function in its ORIGINAL STAGE" (lines 42-44)? 

a2. The subsequent parts (i.e. lines 44-46, lines 54-58) are vague and do not add much context.

a3. "Hence, it is a feasible approach ..." (line 58), may be Authors intended "It is worthwhile"?

a4. The corpus of text from line 61 to line 79 consists of single sentences interleaved with full stops, which makes the manuscript hard to read. Please, rephrase in a more organic way.

Response: a1-4. We have rephrased these sentences in the revised manuscript. Please see line 43-60. 

a5. I would not cite the Warburg effect since it is a bit off-topic for this article (or at least the sentence should be rephrased)

Response: We have removed the sentence in the revised manuscript.

a6. "All in all, glycolysis..." (line 88): please rephrase.

Response: We have rephrased the sentence in the revised manuscript. Please see line 65-66.

Comment 2:Materials and methods

2a. "1% streptomycin/penicillin from solarbio... and 1% antibiotics (Solarbio,..." Please re-check the sentences. Do you put other antibiotics in the medium? Please consider to write the manufacturer's name in the same way throughout the manuscript.

Response: We don’t put other antibiotics in the medium. We have removed the “1% antibiotics (Solarbio, Beijing, China)” and rephrased the sentence in the revised manuscript. Please see line 78-81.

2b. "Mouse against YAP..."(line 129): please rephrase; please cite the antibody that you have used.

Response: We have rephrased the sentence and cited the antibody in the revised manuscript. Please see line 107-109.

Comment 3: Results

3a. Line 191-195: please rephrase, as the sentences are difficult to read.

Response: We have rephrased these sentences in the revised manuscript. Please see line 199-201.

3b. In lines 118-120 Authors said that they conducted three independent bio-repeats, each of which contained three technical replicates: I would expect three different points in the graphs in Figure A. However, the graphs in Figure A have different points in the different conditions (3-5 points per condition). Could the Authors explain in detail how the data that are shown was obtained?

Response: We are sorry for the inaccurate description. We have removed “each of which contained three technical replicates” in the revised manuscript. 

3c. Lines 300-301: "The results confirmed the combination of YAP and 14-3-3n in H9C2 cardiomyocytes.". What does it mean? Please, rephrase.

Response: We have rephrased the sentence in the revised manuscript. Please see line 312-313.

3d. Fig 1, 3: Please clarify "Vector" in the caption. Do you mean negative control (transfection with empty plasmid?)

Response: “Vector” represents negative control. We have explained “Vector” in the revised manuscript. Please see line 97.

---

## [Decision Letter · Decision Letter 1]

9 Jul 2024

Regulation of H9C2 cell hypertrophy by 14-3-3η via inhibiting glycolysis

PONE-D-24-06553R1

Dear Dr. Liu,

We’re pleased to inform you that your manuscript has been judged scientifically suitable for publication and will be formally accepted for publication once it meets all outstanding technical requirements.

Kind regards,

Vincenzo Lionetti, M.D., PhD

Academic Editor

PLOS ONE

Additional Editor Comments (optional):

Reviewers' comments:

Reviewer's Responses to Questions

**Comments to the Author**

1. If the authors have adequately addressed your comments raised in a previous round of review and you feel that this manuscript is now acceptable for publication, you may indicate that here to bypass the “Comments to the Author” section, enter your conflict of interest statement in the “Confidential to Editor” section, and submit your "Accept" recommendation.

Reviewer #2: All comments have been addressed

2. Is the manuscript technically sound, and do the data support the conclusions?

Reviewer #2: Yes

3. Has the statistical analysis been performed appropriately and rigorously? 

Reviewer #2: Yes

4. Have the authors made all data underlying the findings in their manuscript fully available?

Reviewer #2: Yes

5. Is the manuscript presented in an intelligible fashion and written in standard English?

Reviewer #2: Yes

6. Review Comments to the Author

Reviewer #2: Authors addressed all of my concerns and thus, in my opinion, the article is ready for publication.

7. PLOS authors have the option to publish the peer review history of their article (what does this mean?). If published, this will include your full peer review and any attached files.

Reviewer #2: **Yes: **Lorenzo Fontanelli

---

## [Editor Report · Acceptance letter]

11 Jul 2024

PONE-D-24-06553R1 

PLOS ONE

Dear Dr. Liu, 

I'm pleased to inform you that your manuscript has been deemed suitable for publication in PLOS ONE. Congratulations! Your manuscript is now being handed over to our production team.

Kind regards, 

on behalf of

Prof. Vincenzo Lionetti 

Academic Editor

PLOS ONE